# Anastomotic Leak Impact on Long-Term Survival after Right Colectomy for Cancer: A Propensity-Score-Matched Analysis

**DOI:** 10.3390/jcm11154375

**Published:** 2022-07-28

**Authors:** Audrius Dulskas, Justas Kuliavas, Artiomas Sirvys, Augustinas Bausys, Marius Kryzauskas, Klaudija Bickaite, Vilius Abeciunas, Tadas Kaminskas, Tomas Poskus, Kestutis Strupas

**Affiliations:** 1Department of Abdominal and General Surgery and Oncology, National Cancer Institute, Santariskiu Str., LT-08406 Vilnius, Lithuania; justas.kuliavas@gmail.com (J.K.); augustinas.bausys@gmail.com (A.B.); 2Faculty of Medicine, Vilnius University, M. K. Ciurlionio Str. 21, LT-03101 Vilnius, Lithuania; art.sirvys@gmail.com (A.S.); claudia.bickaite@gmail.com (K.B.); vilius.abeciunas@mf.stud.vu.lt (V.A.); tad.kaminskas@gmail.com (T.K.); 3Clinic of Internal Diseases, Family Medicine and Oncology, Institute of Clinical Medicine, Faculty of Medicine, Vilnius University, 2 Santariskiu Street, LT-08661 Vilnius, Lithuania; marius.kryzauskas@santa.lt (M.K.); tomas.poskus@santa.lt (T.P.); kestutis.strupas@santa.lt (K.S.); 4Center of Abdominal Surgery, Vilnius University Hospital Santara Clinics, 2 Santariskiu Street, LT-08661 Vilnius, Lithuania

**Keywords:** anastomotic leak, right sided hemicolectomy, propensity score matching, risk factor, survival rate

## Abstract

Our goal was to assess the impact of anastomotic leaks (ALs) on oncologic outcomes using a case-matched analysis. Patients undergoing right hemicolectomy for cancer between 2014 and 2018 were included. The main variables were the risk factor of anastomotic leak, overall survival and disease-free survival. Propensity score matching was performed according to the patient’s age, co-morbidities and TNM staging as well as the type of procedure. Oncologic outcomes were analyzed. We included 488 patients and performed final analysis on 69 patients. The AL rate was 4.71% (23 patients). Intrahospital mortality was significantly higher in the AL group, at 1.3% (6 of 465) vs. 8.7% (2 of 23), *p* = 0.05. Three-year overall survival (OS) in the non-AL group was higher, although the difference could not be considered significant (71.5% vs. 37.3%, *p* = 0.082); similarly, the likelihood for impaired 3-year progression-free survival (PFS) was lower, but the difference here could also not be considered significant (69.3% vs. 37.3%, *p* = 0.106). Age, advanced tumor stage, lymph node metastases and distant metastases were associated with higher probability of death or recurrence of disease. In contrast, minimally invasive surgery was associated with lower probability of death (HR (95% CI): 0.99 (0.14–0.72); *p* = 0.023) and recurrence of disease (HR (95% CI): 0.94 (0.13–0.68); *p* = 0.020). In an adjusted Cox regression analysis, AL, age and distant metastases were associated with poor long-term survival. Moreover, AL, age and distant metastases were associated with higher probability of recurrence of disease. Based on our results, AL is a significant factor for worse oncologic outcomes. Simple summary: we aimed to assess patients with anastomotic leaks following right hemicolectomy for cancer. These patients were matched to patients without leaks. Propensity score analysis demonstrated that anastomotic leak was a marker of worse oncologic outcomes.

## 1. Introduction

Colorectal cancer (CRC) is one of the most common malignancies worldwide and surgery remains the only potentially curative treatment option for it [1]. Despite recent progress in surgical and anesthesiologic techniques, colorectal resections remain associated with significant postoperative morbidity [2,3]. Anastomotic leakage (AL) is among the most dreadful postoperative complications in modern CRC surgery [4,5,6]. The reported rate of AL varies between 2 and 19% and tends to be higher in patients undergoing surgery for left-side colon or rectal cancer [7,8]. However, leakages of ileocolic anastomoses in right-side colon surgery are also common, with a reported rate of 3.4–8% [9,10,11]. Not only is AL a potentially lethal complication, it may have a negative impact on oncological outcomes as well [2,6,8,12]. The exact mechanisms underlying impaired oncologic outcomes after AL are poorly understood, but several explanations have been proposed. These include the role of the systemic inflammatory response, which promotes the synthesis of oncogenic growth factors and shapes the environment into one suitable for tumor growth [2,13]. Furthermore, AL may lead to extraluminal tumor cell migration and implantation, leading to recurrence and progression of the disease [2,13]. However, most of the evidence on the impact of AL on long-term oncological outcomes has arisen from left-side leakages, thus it remains unclear if ileocolic anastomosis ALs have similar patterns.

Thus, this study aimed to investigate the impact of AL on oncologic outcomes in patients who underwent right colectomy for CRC.

## 2. Materials and Methods

### 2.1. Ethics

The Vilnius Regional Bioethics Committee approved the study (no. 2019/3-116-608). All study-related procedures were performed in accordance with the Declaration of Helsinki of 1964, as revised in 1983. Informed consent was not obtained from the participants because the study was a retrospective investigation.

### 2.2. Patients

This retrospective study was conducted at two major colorectal cancer treatment centers of Lithuania: National Cancer Institute, Vilnius, Lithuania and Vilnius University Hospital Santaros Clinics, Vilnius, Lithuania. All patients who underwent right colectomy for CRC between 2014 and 2018 were included. Patients who underwent emergency surgery or did not receive primary anastomosis were excluded from further analysis. Clinicopathological characteristics and the treatment outcomes of the study patients were extracted from the institutional, prospectively collected databases.

AL was defined as defectiveness at the junction of two ends of intestine with clinically relevant connection between the inside of the intestine, the extraluminal tissue and the abdominal cavity. AL was confirmed by clinical examination (fever, tachycardia, tachypnoea, abdominal tenderness), colonoscopy, radiological evidence of contrast material in the abdominal cavity or ultrasound-guided evidence of perianastomotic fluid with pus or feculent aspirate. Grade A was defined as a leakage that required no active therapeutic intervention. Grade B was defined as a leakage that required active therapeutic intervention but was managed without relaparotomy. Grade C was defined as a leakage that required relaparotomy.

Palliative surgery in this study was defined as a right-colectomy performed for relief of cancer-related symptoms (chronic obstruction or chronic bleeding) in patients with metastatic and incurable cancer.

If incision was used only for the removal of the specimen and anastomosis formation and the preoperatively planned incision was not enlarged (usually 4–6 cm depending on the tumor size), we did not define this as a conversion. If the incision was enlarged for other manipulations, we defined this as a conversion.

### 2.3. Diagnosis, Treatment, and Follow-Up of the Study Patients

The standardized diagnostic pathway of the CRC patients included initial colonoscopy with biopsies followed by chest, abdominal and pelvic computed tomography. Afterward, patients were discussed in a multidisciplinary team meeting and scheduled for right colectomy. Right colectomy was performed using an open or laparascopic approach based on each surgeon’s individual decision. Ileocolic anastomoses were performed side-to-side, end-to-end or end-to-side according to the surgeon’s preference. After patients recovered from surgery, they were allocated to oncologists and received adjuvant chemotherapy if necessary. After all treatment was completed, the patients were followed up with. The standard follow-up protocol consisted of colonoscopy and CT twice a year for the first two years and then annually for the five years after surgery.

### 2.4. Study Outcomes

The primary outcomes of the study were the overall survival (OS) and progression-free survival (PFS). OS was defined as the time from surgery to the patient’s death. PFS was defined as the time from surgery to disease progression or death from any cause. The date of death was obtained from the National Lithuanian Cancer Registry. The secondary outcomes included risk factors for AL after right colectomy.

### 2.5. Statistical Analysis

All statistical analyses were conducted using the statistical program SPSS 25.0 (SPSS, Chicago, IL, USA). Continuous variables were presented as the mean ± standard deviation or median with interquartile range. Categorical variables were calculated as proportions. To minimize the differences between the two groups (AL vs. non-AL), 1:2 propensity score matching (PSM) was performed. Propensity scores were determined by a logistic regression model of covariates using six baseline variables: age, pathological tumor, nodal and metastasis (pTNM) stage, history of stroke and type of radicality of surgery (curative vs. palliative surgery). Such covariates were selected because these variables were found to impact long-term outcomes in this cohort of patients at univariate analysis (data not shown). After propensity scores were calculated, patients in the AL group were matched in a 1:2 ratio with the nearest neighbor from the non-AL group. Overall and disease-free survival rates were analyzed using the Kaplan–Meier method and were compared using the log-rank test. In all statistical analyses, a *p* value of <0.05 was considered to be significant.

The possible risk of bias was decreased by PSM.

## 3. Results

### 3.1. Baseline Characteristics

In total, 488 patients who underwent right colectomy were included in the analysis. The baseline characteristics are presented in Table 1.

AL occurred in 23 (4.71%) patients. One of them (4.3%) had a grade B leakage and twenty-two (95.7%) had grade C leakages. Intrahospital mortality was significantly higher in the AL group (1.3% (6 of 465) vs. 8.7% (2 of 23), *p* = 0.05). For further analysis, these patients were matched with 46 patients without AL by PSM analysis as described above. After PSM, the study groups were well-balanced (Table 2). Postoperative mortality and 3-month readmission rates were similar across the study groups, but hospitalization time in the AL group was significantly longer (25 ± 11 vs. 12 ± 4 days, 0.001).

### 3.2. Long-Term Outcomes

The median follow-up was 34 (Q1:6; Q3:49) months. Three-year OS in the non-AL group was higher, although the difference could not be considered significant (71.5% vs. 37.3%, *p* = 0.082) (Figure 1A). Similarly, we found that patients with ALs had a clear tendency for impaired 3-year PFS, but this difference could not be considered significant either (69.3% vs. 37.3%, *p* = 0.106) (Figure 1B).

Age, advanced tumor stage, lymph node metastases and distant metastases were associated with higher probability of death or recurrence of disease by a univariate Cox regression analysis (Table 3). In contrast, minimally invasive surgery was associated with lower probability of death (HR (95% CI): 0.99 (0.14–0.72); *p* = 0.023) and recurrence of disease (HR (95% CI): 0.94 (0.13–0.68); *p* = 0.020). However, in an adjusted Cox regression analysis, only AL (HR (95% CI): 2.63 (1.26–5.48), *p* = 0.010), age (HR (95% CI): 1.09 (1.03–1.16); *p* = 0.001) and distant metastases (HR (95% CI): 2.34 (1.07–5.10); *p* = 0.032) were associated with poor long-term survival. Further, AL (HR (95% CI): 2.43 (1.18–5.02), *p* = 0.016), age (HR (95% CI): 1.07 (1.02–1.13), *p* = 0.005) and distant metastases (HR (95% CI): 2.27 (1.05–4.90); *p* = 0.036) were associated with higher probability of recurrence as well (Table 4).

## 4. Discussion

Our study shows that AL is a statistically significantly negative prognostic factor in propensity score analysis. We found this to be true in univariate and multivariate analysis. Moreover, age and advanced stage were factors associated with poorer overall survival and disease-free survival. Interestingly, minimally invasive surgery was a factor responsible for better survival.

*Kim* et al. reported significant correlation between AL and vascular diseases [14]. This association can be explained by the necessity of adequate microcirculation for healing the anastomotic site and that patients with histories of cardiovascular or cerebrovascular diseases may have insufficient microcirculation [15].

The anastomosis formation technique (hand-sewn or stapled; end to end, end to side or side to side) had no association with AL (*p* = 0.154 and *p* = 0.945 respectively).

In univariate and multivariate analysis, the independent risk factors of worse OS and PFS were older patient age, advanced cancer stage and distant metastases M1. These factors generally determined a worse general state for a patient, compared to younger patients and patients without metastases; hence, the OS of patients who were older, had a more advanced cancer stage, and distant metastases were shorter. Similar results were seen by Wang et al.: age > 64 years (HR = 1.64 (1.59–1.69), *p* < 0.001) and M1b stage (HR = 1.57 (1.53–1.62), *p* < 0.001) groups had worse OS outcomes, although the OS of N1-2 groups was not negatively impacted (HR = 0.84 (0.81–0.87), *p* < 0.001). Elias et al. also found that older age and N1-2 stages were associated with worse OS (HR = 1.05 (1.04–1.06), *p* < 0.001 for older age and *p* < 0.001 for N1-2 stage) [16,17].

Anastomotic leak is also regarded as a negative prognostic factor for the long-term survival rate after colorectal surgery [7]. However, it is worth mentioning that propensity score matching and randomized controlled trials on this topic are scarce. For example, several studies have found that AL has a statistically significant negative impact on overall survival rate following colorectal surgery [7,8,17]. Moreover, it is important to note that the survival rate graph of the group that experienced anastomotic leak usually differs from the graph of other group mainly and only during the first months following surgery, presumably related to immediate infectious complications and the poor general state of the patient with AL (98% OS in non-AL group and 85% in AL group after 3 months post-surgery according to the OS curve [17]). *Stormark* et al. identified that correlation as well (80–90% OS in AL group and 98–99% OS in non-AL group after first couple of months [7]). *Sueda* et al. performed propensity score matching analysis for overall survival and cancer-specific survival (CSS). Patients were divided into two groups by postoperative intra-abdominal infection due to AL. Authors found that the differences of the OS and CSS curves between the groups were insignificant both before and after matching. The difference between AL and non-AL OS curves was insignificant (*p* = 0.48) before matching, and remained insignificant after matching (*p* = 0.15). A similar result was obtained for CSS: *p* = 0.71 and *p* = 0.72 before and after matching respectively [12]. *Stormark* et al. used a similar method—analysis was performed between non-AL and AL groups, the main variables were relative survival (RS) and conditional relative survival (CRS), and groups were compared only one year after the surgery, thus eliminating negative outcomes due to the immediate infectious complications of AL. AL had a significant negative influence on RS in all cancer stages, but the impact of AL on CRS was significant only for those with stage III disease; consequently, these results are partly comparable to ours. *Koedam* et al. performed an analysis of the relation between AL and DFS without patient matching. AL was found to be an insignificant factor of decreased DFS (HR 1.40, 95% CI 0.69–2.84). *Voron* et al. found that AL had a significant negative impact on both OS and DFS in non-matched analysis (*p* < 0.001 for OS and *p* = 0.003 for DFS).

In summary, the role of AL as a risk factor of decreased OS and DFS in the literature is controversial, and our results support the opinion that AL is a marker of worse oncological outcome.

AL can worsen oncological outcomes by three main mechanisms. Firstly, anastomotic leak leads to the inflammation of nearby tissues that become a favorable environment for intraluminal cancer cells, which are clones of primary tumor cells, to implant themselves in [18,19]; consequently, local tumor recurrence determines worse oncological outcomes. The second mechanism is related to metachronous carcinogenesis, which is when primary tumor microenvironmental changes in the anastomotic site in the case of AL lead to genetic instability and secondary tumor growth [20]. The third mechanism supports the significance of acute-phase mediators in cancer biology. Inflammatory biomarkers (such as TNF-α, IL-1, IL-6, vascular endothelial growth factor, matrix metalloproteinases) may lead to metastases, tumor progression and inefficiency of adjuvant chemotherapy [19,20,21]. An additional mechanism is hypothesized that suggests the role of AL-induced inflammation on circulating cancer cells. Inflammation continuously implicates inflammatory cells to the AL site, and that initiates a cascade of immune cell reactions and processes like angiogenesis, wound repair and cell proliferation; thus, the anastomotic site becomes a fecund medium for circulating cancer cells to implant and contribute to local recurrence and worse OS [21,22]. Other consequences of AL also include postponed adjuvant chemotherapy; worse nutritional status; longer hospital stay, which also contributes to longer contact with hospital infections; stress; and the requiring of longer antibiotic therapy or additional invasive procedures [22]. These factors may lead the intestinal microbiome to change to pathogenic, consequently progressing inflammation in the anastomotic site and raising the probability of cancer cell implantation (especially due to postponed adjuvant chemotherapy) [22,23]. The aforementioned mechanisms clearly show that AL should have a poor impact on oncologic outcomes and OS. In contrast, *Fransgaard* et al. found that postoperative complications tended to delay adjuvant chemotherapy (OR = 4.56, 3.67–5.66 95% CI, *p* < 0.001) and were not associated with worse DFS (HR = 1.02, 0.88–1.18 95% CI, *p* = 0.8), recurrence-free survival (HR = 1.05, 0.89–1.25 95% CI, *p* = 0.56) or with higher mortality rate (HR = 1.04, 0.86–1.26 95% CI, *p* = 0.67). Moreover, *Bashir M* et al. found in their meta-analysis that anastomotic leak had a statistically insignificant influence on local recurrence rate (7.5% local recurrence in AL group and 6% in another group, (RR) 1.16 (95% CI 0.84–1.59)) [2]. Our study results corroborate the findings of the meta-analysis by *Bashir M* et al., since the recurrence rate in the non-AL group was 0.2% compared to 0% in the AL group, *p* = 1.00. The difference in the distant recurrence rate according to the results of *Bashir M* et al. appears to be insignificant as well between the groups (25% and 12%, AL group and another group respectively, RR = 1.44 (95% CI 0.52–3.96, I^2^ = 97%, *p* = 0.48)) [2]. We also found that the rate of distant recurrence in non-AL and AL groups was similar and that neither of the distant recurrent sites had a statistically significant difference between the groups. Thus, based on our results, AL has no impact on local or distant recurrence of colon cancer in a case-matched cohort.

Obviously, our study had several limitations. First, this was a single-center study with a relatively low sample size. However, a multicenter approach and significant national-registry-based long-term follow-ups would increase the power of the study to demonstrate that AL might be associated with impaired long-term outcomes in patients undergoing surgery for right-sided CRC.

## 5. Conclusions

Based on our results, AL was a significant factor for worse oncologic outcomes. Still, further larger studies are needed on this topic to provide stronger evidence.

## Figures and Tables

**Figure 1 jcm-11-04375-f001:**
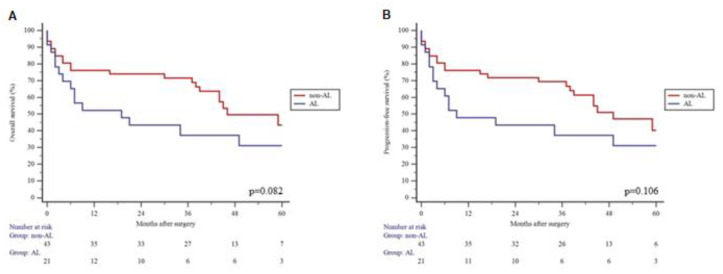
3-year overall survival (**A**) and progression-free survival (**B**) in anastomotic leak (AL) group vs. no leak (non-AL) group.

**Table 1 jcm-11-04375-t001:** Baseline characteristics of the study patients.

**Age, Years; Mean ± SD**		69.3 ± 11.0
**Gender; *n* (%)**	Male	206 (42.2%)
	Female	282 (57.8%)
ASA score; *n* (%)	I–II	208 (42.6%)
	III–IV	280 (57.4%)
Chronic kidney failure; *n* (%)		10 (2%)
Diabetes; *n* (%)		65 (13.3%)
Coronary heart; *n* (%)		130 (26.6%)
History of stroke; *n* (%)		15 (3.1%)
Tumor localization; *n* (%)	Caecum; *n* (%)	128 (26.2%)
	Ascending colon; *n* (%)	265 (54.3%)
	Hepatic flexure; *n* (%)	63 (12.9%)
	Transverse colon; *n* (%)	32 (6.6%)
Surgical radicality	Radical; *n* (%)	453 (92.8%)
	Palliative; *n* (%)	35 (7.2%)
pT	T1/2; *n* (%)	69 (14.1%)
	T3/4; *n* (%)	419 (85.9%)
pN	N0; *n* (%)	249 (51.0%)
	N+; n (%)	239 (49.0%)
pM	0; *n* (%)	415 (85.0%)
	1; *n* (%)	73 (15.0%)
Surgical approach	Open; *n* (%)	402 (82.4%)
	MIS; *n* (%)	86 (17.6%)
Type of anastomosis	End-to-end; *n* (%)	43 (8.8%)
	End-to-side; *n* (%)	204 (41.8%)
	Side-to-side; *n* (%)	241 (49.4%)
Anastomotic technique	Hand sewn; *n* (%)	484 (99.2%)
	Stapled; *n* (%)	4 (0.8%)
Postoperative complications; *n* (%)		111 (22.7%)
Anastomotic leakage; *n* (%)		23 (4.7%)
Intrahospital mortality; *n* (%)		8 (1.6%)
Postoperative hospitalization length, days; mean ± SD		12 ± 7
3 months readmission rate; *n* (%)		32 (6.6%)

**Table 2 jcm-11-04375-t002:** Baseline characteristics of patients in the anastomotic leakage and non-anastomotic leakage groups before and after propensity score matching.

		Before Propensity Score Matching			After Propensity Score Matching		
		AL (*n* = 23)	Non-AL (*n* = 465)	*p* Value	AL (*n* = 23)	Non-AL (*n* = 46)	*p* Value
**Age, years; mean ± SD**		72 ± 10	69 ± 11	0.227	72 ± 10	73 ± 8	0.584
Gender; *n* (%)	Male	9 (39.1%)	197 (42.4%)	0.759	9 (39.1%)	20 (43.5%)	0.730
	Female	14 (60.9%)	268 (57.6%)		14 (60.9%)	26 (56.5%)	
ASA score; *n* (%)	I-II	8 (34.8%)	200 (43.0%)	0.436	8 (34.8%)	13 (28.3%)	0.579
	III-IV	15 (65.2%)	265 (57.0%)		15 (65.2%)	33 (71.7%)	
Chronic kidney failure; *n* (%)		1 (4.3%)	9 (1.9%)	0.425	1 (4.3%)	0 (0%)	0.333
Diabetes; *n* (%)		2 (8.7%)	63 (13.5%)	0.504	2 (8.7%)	8 (17.4%)	0.477
Coronary heart; *n* (%)		8 (34.8%)	122 (26.2%)	0.365	8 (34.8%)	14 (30.4%)	0.715
History of stroke; *n* (%)		3 (13.0%)	12 (2.6%)	0.005	3 (13.0%)	6 (13.0%)	0.999
Tumor localization; *n* (%)	Caecum; *n* (%)	7 (30.4%)	127 (27.3%)	0.445	7 (30.4%)	14 (30.4%)	0.974
	Ascending colon; *n* (%)	11 (47.8%)	248 (53.3%)		11 (47.8%)	24 (52.2%)	
	Hepatic flexure; *n* (%)	2 (8.7%)	64 (13.8%)		2 (8.7%)	3 (6.5%)	
	Transverse colon; *n* (%)	3 (13.1%)	26 (5.6%)		3 (13.1%)	5 (10.9%)	
Surgical radicality	Radical; *n* (%)	20 (87.0%)	433 (93.1%)	0.264	20 (87.0%)	38 (82.6%)	0.740
	Palliative; *n* (%)	3 (13.0%)	32 (6.9%)		3 (13.0%)	8 (17.4%)	
pT	T1/2; *n* (%)	3 (13.0%)	66 (14.2%)	0.877	3 (13.0%)	8 (17.4%)	0.740
	T3/4; *n* (%)	20 (87.0%)	399 (85.8%)		20 (87.0%)	38 (82.6%)	
pN	N0; *n* (%)	14 (60.9%)	235 (50.5%)	0.333	14 (60.9%)	24 (52.2%)	0.494
	N+; *n* (%)	9 (39.1%)	230 (49.5%)		9 (39.1%)	22 (47.8%)	
pM	0; *n* (%)	18 (78.3%)	397 (85.4%)	0.350	18 (78.3%)	36 (78.3%)	0.999
	1; *n* (%)	5 (21.7%)	68 (14.6%)		5 (21.7%)	10 (21.7%)	
Surgical approach	Open; *n* (%)	20 (87.0%)	382 (82.2%)	0.555	20 (87.0%)	37 (80.4%)	0.500
	MIS; *n* (%)	3 (13.0%)	83 (17.8%)		3 (13.0%)	9 (19.6%)	
Type of anastomosis	End-to-end; *n* (%)	2 (8.7%)	41 (8.8%)	0.961	2 (8.7%)	3 (6.5%)	0.945
	End-to-side; *n* (%)	9 (39.1%)	195 (41.9%)		9 (39.1%)	18 (39.1%)	
	Side-to-side; *n* (%)	12 (52.2%)	229 (49.3%)		12 (52.2%)	25 (54.3%)	
Anastomotic technique	Hand sewn; *n* (%)	22 (95.7%)	462 (99.4%)	0.055	22 (95.7%)	46 (100%)	0.154
	Stapled; *n* (%)	1 (4.3%)	3 (0.6%)		1 (4.3%)	0 (0%)	
Intrahospital mortality; *n* (%)		2 (8.7%)	6 (1.3%)	0.006	2 (8.7%)	1 (2.2%)	0.210
Postoperative hospitalization length, days; mean ± SD		25 ± 11	12 ± 7	0.001	25 ± 11	12 ± 4	0.001
3 months readmission rate; *n* (%)		5 (21.7%)	27 (5.8%)	0.003	5 (21.7%)	4 (8.7%)	0.129

**Table 3 jcm-11-04375-t003:** Univariate Cox regression analysis for overall and disease-free survival in the propensity-score-matched cohort.

*Variable*	*Category*	*Overall Survival*		*Disease-Free Survival*	
		*HR (95% CI)*	*p Value*	*HR (95% CI)*	*p Value*
**Anastomotic leakage**	No	1 (Reference)		1 (Reference)	
	Yes	1.77 (0.91–3.46)	*0.091*	1.69 (0.87–3.28)	*0.116*
Age		1.09 (1.04–1.14)	*0.001*	1.08 (1.03–1.13)	*0.001*
Gender	Male	1 (Reference)		1 (Reference)	
	Female	1.58 (0.79–3.16)	*0.196*	1.70 (0.85–3.38)	*0.130*
ASA score	1–2	1 (Reference)		1 (Reference)	
	3–4	1.72 (0.78–3.79)	*0.173*	1.77 (0.81–3.89)	*0.150*
pT	1–2	1 (Reference)		1 (Reference)	
	3–4	4.00 (0.96–16.68)	*0.05*	4.21 (1.01–17.54)	*0.048*
pN	N0	1 (Reference)		1 (Reference)	
	N+	2.31 (1.17–4.52)	*0.015*	2.23 (1.15–4.33)	*0.017*
pM	M0	1 (Reference)		1 (Reference)	
	M1	3.84 (1.89–7.80)	*0.001*	3.66 (1.82–7.39)	*0.001*
Type of surgery	Open	1 (Reference)		1 (Reference)	
	MIS	0.99 (0.14–0.72)	*0.023*	0.94 (0.13–0.68)	*0.020*

MIS: minimally invasive surgery.

**Table 4 jcm-11-04375-t004:** Multivariable Cox regression analysis for overall and disease-free survival in the propensity-score-matched cohort.

*Variable*	*Category*	*Overall Survival*		*Disease-Free Survival*	
		*HR (95% CI)*	*p Value*	*HR (95% CI)*	*p Value*
**Anastomotic leakage**	No	1 (Reference)		1 (Reference)	
	Yes	2.63 (1.26–5.48)	*0.010*	2.43 (1.18–5.02)	*0.016*
Age		1.09 (1.03–1.16)	*0.001*	1.07 (1.02–1.13)	*0.005*
Gender	Male	1 (Reference)		1 (Reference)	
	Female	1.77 (0.81–3.82)	*0.146*	1.96 (0.91–4.21)	*0.084*
ASA score	1–2	1 (Reference)		1 (Reference)	
	3–4	0.87 (0.36–2.13)	*0.771*	0.83 (0.34–2.04)	*0.697*
pT	1–2	1 (Reference)		1 (Reference)	
	3–4	1.36 (0.27–6.76)	*0.704*	1.67 (0.34–8.12)	*0.524*
pN	N0	1 (Reference)		1 (Reference)	
	N+	1.60 (0.70–3.66)	*0.259*	1.48 (0.66–3.32)	*0.342*
pM	M0	1 (Reference)		1 (Reference)	
	M1	2.34 (1.07–5.10)	*0.032*	2.27 (1.05–4.90)	*0.036*
Type of surgery	Open	1 (Reference)		1 (Reference)	
	MIS	6.79 (0.87–52.76)	*0.067*	7.07 (0.92–54.15)	*0.059*

## Data Availability

Data are contained within the article. The data presented in this study are available in the article.

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
