# Peer review of "Anastomotic Leak Impact on Long-Term Survival after Right Colectomy for Cancer: A Propensity-Score-Matched Analysis"

_jcm, 2022, doi:10.3390/jcm11154375_

Round 1
Reviewer 1 Report
I had the pleasure to review the paper: Anastomotic leak impact on long-term survival after right colectomy for cancer: a propensity scores matched analysis. Anastomotic leak impact on survival.
I have the following observations:
1. Title is cancer, not caner (minor typo).
2. Abstract - change the number of the patients "488 patients included" - not quite true, there were 69 pts included, 488 were analyzed at the first stage. and specified something about the methods in the abstract.
3. It is strange to mix radical surgery and palliative surgery patients ( at least define what palliative means), i would be more specifique or remove these patients from the analysis group, if there was no emergency (according with exclusion criteria) with these patients why performing surgery in the first place? did you match the liver metastasis volume in these patients too? from my point of view, you cannot mix these patients, but I am not the Editor!
4. Laparoscopy comparison 3 versus 9 patients? why you did that? any comment is irrelevant at this numbers so please remove the discussion part about laparoscopy, or any conclusion based on these numbers.
4.1. You performed the anastomosis open in laparoscopic cases (4 stapled from 488) - what is the limit of the incision to declare conversion?
5. Nothing about limphnodes ( a surrogate for radicality) but if you didnt have evaluation of complete mesocolon excision from pathology is the best we have!
6. Nothing about hte biases in your paper, maybe you can re-write the paper according with strobe check-list for retrospective studies?
Author Response
A point-by-point response to the reviewers and editor's comments:
Dear Editor,
Thank you for your letter and constructive comments concerning our manuscript entitled “Anastomotic leak impact on long-term survival after right co-lectomy for cancer: a propensity scores matched analysis. Anastomotic leak impact on survival”. The paper was revised substantially. Following changes have been made. They are as follows:
Reviewer #1:
- Title is cancer, not caner (minor typo).
Changed, thank you for the spelling.
- Abstract - change the number of the patients "488 patients included" - not quite true, there were 69 pts included, 488 were analyzed at the first stage. and specified something about the methods in the abstract.
We have changed the numbers as per suggestion.
- It is strange to mix radical surgery and palliative surgery patients ( at least define what palliative means), i would be more specifique or remove these patients from the analysis group, if there was no emergency (according with exclusion criteria) with these patients why performing surgery in the first place? did you match the liver metastasis volume in these patients too? from my point of view, you cannot mix these patients, but I am not the Editor!
We agree with the reviewer, that patients undergoing palliative and radical surgery differ. As suggested, we defined palliative procedures: “Palliative surgery in this study was defined as a right-colectomy performed for relief of cancer related symptoms (obstruction or chronic bleeding) in patients with metastatic and incurable cancer”. We would like to explain that we started with a surgery-first approach if patients had a metastatic and incurable cancer but suffered tumor related symptoms – chronic obstruction or chronic bleeding. By propensity score matching we adjusted both groups to this potential co-founder, thus we did not consider excluding these patients.
- Laparoscopy comparison 3 versus 9 patients? why you did that? any comment is irrelevant at this numbers so please remove the discussion part about laparoscopy, or any conclusion based on these numbers.
This is a correct notification. We have deleted this part in the discussion and corrected the conclusions
- You performed the anastomosis open in laparoscopic cases (4 stapled from 488) - what is the limit of the incision to declare conversion?
If incision was used only for the removal of the specimen and anastomosis formation and the planned preoperatively incision was not increased (usually it is 4-6 cm depending on the tumor size). If incision was increased for other manipulations – we defined this as a conversion. This was included in the Methods part.
- Nothing about limphnodes ( a surrogate for radicality) but if you didnt have evaluation of complete mesocolon excision from pathology is the best we have!
We agree with the reviewer, that evaluation of CME is an important parameter in the pathology report. Although, this study is retrospective, and at time the patients underwent surgery evaluation of CME was not included in our institutions pathology reports. The specimens were harvested for pathology evaluation, thus we cannot add this parameter to study outcomes.
- Nothing about hte biases in your paper, maybe you can re-write the paper according with strobe check-list for retrospective studies?
Strobe checklist was added.
The manuscript improved a lot.
Thank you for the comments and suggestions.
Sincerely
Audrius Dulskas
MD, PhD
Reviewer 2 Report
Thank you for your effort to enhance our knowledge of the care of colorectal cancer patients.
The authors demonstrated the importance of anastomosis leakage for oncologic outcomes in patients with right-sided colon cancer.
However, I think there are some technical issues to clarify the results and the conclusion.
1. In the method section, the authors defined the overall and disease-free survival. Why did you include 'death' to calculate the DFS? As we know, regardless of survival or death, a recurrence has to be a risk factor for calculating DFS.
2. In statistical analysis, the authors demonstrated six baseline variables for PSM. Can you explain why the history of stroke of patients' underlying diseases was included in baseline variables? In table 1, there were some potential risk factors (coronary heart disease, DM...) related to patients' vascular condition. I think this issue has to be explained by the authors.
3. In the results, the authors described the grade of anastomosis leakage. However, there was no description of the grading system for anastomosis leakage in this manuscript. So, I would like to recommend the authors add the grading system they used into the method section.
4. In table 2, the authors presented results after PSM. However, for clarifying the results, the authors have to present both the results from 'before PSM' and 'after PSM'. In other words, any changes in results through PSM in this table have to be presented.
5. In table 1 and 2, the patients with metastatic colorectal cancer (stage IV) or those who underwent palliative resection were included. As I know, stage IV disease is already a systemic disease. So, I think it is not appropriate in this manuscript to identify the relationship between anastomosis leakage and oncologic outcomes. So, the authors had better analyze the data again.
6. The authors demonstrated the type of anastomosis as a risk factor related to surgical technique. However, there are lots of risk factors of anastomosis leakage related to oncologic surgical techniques including the range of lymph node dissection (D1, D2, or D3), and the location of feeding vessel ligation (especially, which branch of the mid colic artery was ligated), and so on. I think when the authors control these variables related to technical factors affecting oncologic outcomes the anastomosis leakage can be one of the important oncologic risk factors in this study.
Once again, I appreciate the authors' effort to advance our knowledge to care the patients with colorectal cancer.
Author Response
A point-by-point response to the reviewers and editor's comments:
Dear Editor,
Thank you for your letter and constructive comments concerning our manuscript entitled “Anastomotic leak impact on long-term survival after right colectomy for cancer: a propensity scores matched analysis. Anastomotic leak impact on survival”. The paper was revised substantially. Following changes have been made. They are as follows:
- In the method section, the authors defined the overall and disease-free survival. Why did you include 'death' to calculate the DFS? As we know, regardless of survival or death, a recurrence has to be a risk factor for calculating DFS. We agree with the reviewer, that definition of DFS may vary. Although, we used the most common definition which defined DFS as: “DFS is generally defined as the time from randomization until tumor recurrence or any-cause death after treatments given with curative intent.” (Fiteni et al., 2014; PMID: 24440056)
- In statistical analysis, the authors demonstrated six baseline variables for PSM. Can you explain why the history of stroke of patients' underlying diseases was included in baseline variables? In table 1, there were some potential risk factors (coronary heart disease, DM...) related to patients' vascular condition. I think this issue has to be explained by the authors. As suggested we explained the rationale for these covariates selection: “Such covariates were selected because these variables were found to have impact on long-term outcomes in this cohort of patients at univariate analysis (data not shown).”
- In the results, the authors described the grade of anastomosis leakage. However, there was no description of the grading system for anastomosis leakage in this manuscript. So, I would like to recommend the authors add the grading system they used into the method section.
Grading system was added to the Methods section. Thank you.
- In table 2, the authors presented results after PSM. However, for clarifying the results, the authors have to present both the results from 'before PSM' and 'after PSM'. In other words, any changes in results through PSM in this table have to be presented.
As suggested Table 2 was supplemented with result before PSM.
- In table 1 and 2, the patients with metastatic colorectal cancer (stage IV) or those who underwent palliative resection were included. As I know, stage IV disease is already a systemic disease. So, I think it is not appropriate in this manuscript to identify the relationship between anastomosis leakage and oncologic outcomes. So, the authors had better analyze the data again.
We agree with the reviewer, that patients undergoing palliative and radical surgery differ same as patients with systemic and locoregional disease. Although, stage IV patients requiring right-colectomy represent real-world situation. Further, these patients are at particularly high risk for leakage, and they struggle to recover from it. Therefore, aiming to better represent real-world data, we choose to not exclude stage IV patients, but rather adjust both groups for this potential co-founder. It was successfully done by PSM analysis and both groups are homogenous according to M status or palliative procedures rate.
- The authors demonstrated the type of anastomosis as a risk factor related to surgical technique. However, there are lots of risk factors of anastomosis leakage related to oncologic surgical techniques including the range of lymph node dissection (D1, D2, or D3), and the location of feeding vessel ligation (especially, which branch of the mid colic artery was ligated), and so on. I think when the authors control these variables related to technical factors affecting oncologic outcomes the anastomosis leakage can be one of the important oncologic risk factors in this study.
Thank you for the comment. All our patients undergo standardized surgery – meaning CME and high vascular ligation (or so called D3) and ligation of the right branch of the middle colic (except for the extended right hemicolectomy where trunk of middle colic artery is ligated).
The manuscript improved a lot.
Thank you for the comments and suggestions.
Sincerely
Audrius Dulskas
MD, PhD
Round 2
Reviewer 1 Report
Dear Authors, still no limitations talk about your study, but this is up to the Editor to ask. (Strobe guidelines)
maybe I am too technical, but I would move to methods from the statistical analysis the part about palliative surgery, is about defining variables, not about statistics. (Strobe guidelines)
Author Response
Dear Reviewer
Thank you for you comments!
Dear Authors, still no limitations talk about your study, but this is up to the Editor to ask. (Strobe guidelines)
We appreciate your comment.
maybe I am too technical, but I would move to methods from the statistical analysis the part about palliative surgery, is about defining variables, not about statistics. (Strobe guidelines)
We totally agree – it was moved.
Sincerely
Audrius Dulskas, MD PhD
Reviewer 2 Report
Thank you for your nice revision and reply.
However, it is not easy to agree with your answer for Q1, Q2, Q5, and Q6.
Especially, the definition of DFS and the inclusion of stage IV appear to be confounding factors in the result and the conclusion of this manuscript.
I would like to reconsider the method to draw a clear conclusion.
Thank you for your efforts.